# A Deacetylase *Cq*SIRT1 Promotes WSSV Infection by Binding to Viral Envelope Proteins in *Cherax quadricarinatus*

**DOI:** 10.3390/v14081733

**Published:** 2022-08-06

**Authors:** Shucheng Zheng, Fanjuan Meng, Dongli Li, Lingke Liu, Di Ge, Qing Wang, Haipeng Liu

**Affiliations:** 1State Key Laboratory of Marine Environmental Science, State-Province Joint Engineering Laboratory of Marine Bioproducts and Technology, College of Ocean and Earth Sciences, Xiamen University, Xiamen 361102, China; 2Laboratory for Marine Biology and Biotechnology, Pilot National Laboratory for Marine Science and Technology, Qingdao 266100, China; 3Key Laboratory of Fishery Drug Development of Ministry of Agriculture, Key Laboratory of Aquatic Animal Immune Technology of Guangdong Province, Pearl River Fisheries Research Institute, Chinese Academy of Fishery Sciences, Guangzhou 510380, China

**Keywords:** SIRT1, lysine deacetylases, white spot syndrome virus, envelope proteins, *Cherax quadricarinatus*

## Abstract

Sirtuin 1 (SIRT1), a member of the class III lysine deacetylases, exhibits powerful functional diversity in physiological processes and disease occurrences. However, the potential molecular mechanism underlying the role of SIRT1 during viral infection in crustaceans is poorly understood. Herein, SIRT1 was functionally characterized from the red claw crayfish *Cherax quadricarinatus*, which possesses typically conserved deacetylase domains and strong evolutionary relationships across various species. Moreover, gene knockdown of *Cq*SIRT1 in crayfish haematopoietic tissue (Hpt) cell culture inhibited white spot syndrome virus (WSSV) late envelope gene *vp28* transcription. In contrast, enhancement of deacetylase activity using a pharmacological activator promoted the replication of WSSV. Mechanically, *Cq*SIRT1 was co-localized with viral envelope protein VP28 in the nuclei of Hpt cells and directly bound to VP28 with protein pulldown and co-immunoprecipitation assays. Furthermore, *Cq*SIRT1 also interacted with another two viral envelope proteins, VP24 and VP26. To the best of our knowledge, this is the first report that WSSV structural proteins are linked to lysine deacetylases, providing a better understanding of the role of *Cq*SIRT1 during WSSV infection and novel insights into the basic mechanism underlying the function of lysine deacetylases in crustaceans.

## 1. Introduction

Lysine acetylation is an essential and evolutionarily conserved post-translation modification that is relevant to numerous cellular events and human disease. In general, acetylation is strictly controlled by lysine acetyltransferases (KATs) through the transfer of an acetyl group from acetyl-CoA to the ε-amino side chain of lysine, and it is reversed by lysine deacetylases (KDACs) [1]. KDACs have phylogenetically been divided into two major families: Zn^2+^-dependent histone deacetylases (HDACs) and nicotinamide adenine dinucleotide (NAD^+^)-dependent sirtuin deacetylases (SIRTs). The seven members of the SIRT family (SIRT1–SIRT7) are ubiquitously expressed in a variety of cell and tissue types, and they are dependent on NAD^+^ as a cofactor and modulate various cellular functions, including transcriptional regulation, cell proliferation, metabolism and immune response [2]. In particular, SIRT1, as the best characterized member, is directly linked to more than 40% of the acetylation sites of cellular proteome and has many physiological effects, such as on longevity and disease, through deacetylation of target substrates [3].

SIRT1 exhibits broad-range antiviral properties against several DNA viruses, including human cytomegalovirus and herpes simplex virus-1, and RNA viruses, such as influenza A virus, as well as bacteriophages [4]. However, several studies have also demonstrated that SIRT1 can be hijacked by viruses to modulate viral or host transcription machineries for the benefit of viral infection, depending on the virus type. For instance, SIRT1 mediates the latency of Kaposi’s sarcoma-associated herpesvirus (KSHV) by binding to the master lytic trans-activator RTA (ORF50) of KSHV and inhibiting RTA’s transactivation function to prevent the expression of its downstream genes, enabling the virus to survive and escape from the immune surveillance of host [5]. In contrast, SIRT1 can activate hepatitis B virus (HBV) transcription by promoting the recruitment of c-Jun transcription factor to precore (preC) promoter of HBV and binding to HBV X transactivator (HBX) [6,7]. Another mechanism by which SIRT1 assists viral infection is its recruitment by a virus to the viral genome or replication complex. Typically, HBV gene transcription is activated via recruitment of SIRT1 to covalently closed circular DNA (cccDNA) followed by deacetylation of several transcriptional regulators. Likewise, SIRT1, as a member of the human papillomavirus helicase E1-E2 replication complex, is recruited to the viral origin of replication in an E1-E2-dependent manner [8,9]. These findings strongly indicate that SIRT1 is a key cellular deacetylase linked to viral infection through the targeting of a number of viral or cellular substrates. However, most studies on the role of SIRT1 were associated with human disease, and the function of SIRT1 during viral infection in crustaceans was rarely reported.

In crustaceans, white spot syndrome virus (WSSV) is the most fatally viral pathogen involving envelope proteins and the double-strand DNA genome, and it has resulted in huge economic losses in China and Southeast Asia [10]. Though a growing number of studies have been conducted on the etiology of WSSV and the modulatory mechanism underlying host immune defense, the fundamental mechanisms involved in lysine deacetylation mediated by KDACs have only rarely been reported [11,12,13,14]. Previously, a study found that SIRT1 from *Penaeus vannamei* could promote WSSV infection, but the potential molecular mechanism remains to be fully characterized [15]. In our present study, we found that *Cq*SIRT1 from red claw crayfish *Cherax quadricarinatus* (*C. quadricarinatus*) promotes WSSV infection with deacetylase activity through binding to several viral envelope proteins. These findings will help in better understanding the pathogenic mechanism of WSSV and provide a snapshot about the broad modulatory function of *Cq*SIRT1 in crustaceans.

## 2. Materials and Methods

### 2.1. Animals, Hpt Cell Cultures and WSSV

Mature freshwater red claw crayfish *C. quadricarinatus* were purchased from Shunxing Aquaculture Co., Ltd., Longyan, Fujian Province, China, and cultivated in aerated tanks at 26 °C for at least two weeks before use in experiments. Commercial feed was provided once a week (Haid Group, Guangzhou, China). Hpt cells were isolated from the male *C. quadricarinatus* and cultured as described by Söderhäll et al. [16]. Briefly, the Hpt was dissected from the dorsal side of the stomach and incubated in 300 μL of 0.1% collagenase (type I) and 0.1% collagenase (type IV) in crayfish phosphate-buffered saline (CPBS) (10 mM Na_2_HPO_4_, 10 mM KH_2_PO_4_, 0.15 M NaCl, 10 mM CaCl_2_, 10 mM MnCl_2_, 2.7 mM KCl; pH 6.8) at room temperature for 45 min. Then, the tissue was gently passed 10–20 times through a Pasteur pipette and centrifuged at 2000× *g* for 5 min to obtain a cell pellet. The pellet was washed once in 500 μL of CPBS with centrifugation at 2000× *g* for 5 min and resuspended in 500 μL of L-15 medium (HyClone, Logan, UT, USA).

WSSV was prepared and purified as described by Xie et al. and quantified via absolute quantification with PCR [17]. Briefly, WSSV-infected tissue from crayfish *Procambarus clarkia* was homogenized in 500 mL TNE buffer (50 mM Tris–HCl, 400 mM NaCl, 5 mM EDTA; pH 8.5) supplemented with protease inhibitors and then centrifuged at 3500× *g* for 5 min at 4 °C. After filtering with a nylon net (400 mesh), the supernatant was centrifuged at 30,000× *g* for 30 min at 4 °C and the lower white pellet was suspended in 10 mL TM buffer (50 mM Tris–HCl, 10 mM MgCl_2_; pH 7.5). The virus particles were sedimented with centrifugation at 30,000× *g* for 20 min at 4 °C, and then resuspended in 1 mL TM buffer containing 0.1% NaN_3_. The virions were observed with transmission electron microscopy.

### 2.2. RNA Extraction and Reverse Transcription

Total RNA from Hpt cell cultures was extracted using a Spin Column Animal Total RNA Purification Kit (Sangon Biotech, Shanghai, China) according to the manufacturer’s protocol. The integrality of RNA was analyzed using 1.0% agarose electrophoresis and the concentration was measured with a NanoDrop 2000 spectrophotometer (Thermo Scientific, Waltham, Massachusetts, USA). One microgram of total RNA was reversibly transcribed to cDNA and genomic DNA was removed using HiScript^®^ II Q RT SuperMix for qPCR (+gDNA wiper) (Vazyme, Nanjing, China) according to the manufacturer’s instruction.

### 2.3. Gene Cloning and Molecular Characteristics Analysis

The primer pairs (Table 1) were designed based on the sequence obtained from a transcriptome library of *C. quadricarinatus* in our lab. The open reading frame (ORF) of the gene was cloned using PrimeSTAR^®^ Max DNA Polymerase (Takara, Kusatsu, Japan) according to the manufacturer’s instructions. The PCR conditions were as follows: 35 cycles of 94 °C for 10 s, 55 °C for 5 s and 72 °C for 15 s. Amplified PCR products were purified using a Gel Extraction Kit (Sangon Biotech, Shanghai, China) and ligated into a pMD18-T vector (Takara, Kusatsu, Japan) for sequencing (Sangon Biotech, Shanghai, China).

Sequence analysis was performed using DNAstar and the Basic Local Alignment Search Tool (BLAST) program. The conserved domains were analyzed in SMART (http://smart.embl-heidelberg.de, accessed on 15 June 2022). Multiple sequence alignment was performed using DNAMAN software. The phylogenetic tree was constructed with MEGA X using the neighbor-joining method based on the deduced complete acid amino sequence [18].

### 2.4. Semi-Quantitative RT-PCR and Real-Time Fluorescent Quantitative PCR

Semi-quantitative RT-PCR was performed using 2 × Taq Master Mix (Dye Plus) (Vazyme, Nanjing, China). The primer pairs are shown in Table 1. The PCR conditions were as follows: 95 °C for 5 min; 30 cycles of 95 °C for 15 s, 58 °C for 15 s and 72 °C for 30 s; 72 °C for 5 min. All amplified products were analyzed with 1.5% agarose electrophoresis. The 16s ribosome RNA gene (GenBank no: AF135975.1) of red claw crayfish was used as an internal standard.

Real-time fluorescent quantitative PCR (qRT-PCR) was performed using ChamQ Universal SYBR qPCR Master Mix (Vazyme, Nanjing, China). The primer pairs are shown in Table 1. The reaction volume consisted of 20 μL of mixture containing 10 μL of SYBR Green Master (2×), 0.4 μL of each primer (10 mM), 1.0 μL of cDNA and up to 20 μL of sterile water. The qRT-PCR program was as follows: 95 °C for 30 s; 40 cycles of 95 °C for 10 s and 60 °C for 35 s. The amplification specificity of each primer pair was determined by analyzing the melt curves following PCR reactions. The 16s ribosome RNA gene was used as an internal standard. All samples were analyzed in triplicate and the expression level of the targeted gene relative to 16s ribosomal gene expression was quantified using the 2^−△△^Ct method [19].

### 2.5. Double-Strand RNA Production and RNA Interference

The primers for PCR amplification of the target sequence are shown in Table 1. The dsRNA was synthesized using a RiboMAX™ Large Scale RNA Production Systems-SP6 and T7 kit (Promega, Madison, WI, USA) according to the manufacturer’s instructions and purified with the Trizol Isolation Reagent (Invitrogen, Waltham, MA, USA). An RNA interference assay was carried out with double-strand RNA (dsRNA) transfection into Hpt cell cultures. Hpt cells (5 × 10^5^ cells) were seeded into 24-well culture plates and cultured in 500 μL of L15 medium (HyClone, Logan, UT, USA) with minor modifications. Four hundred nanograms of dsRNA was transfected with 0.8 μL of Cellfectin II Reagent (Invitrogen, Waltham, MA, USA) in Hpt cells according to the manufacturer’s protocol. After 36 h, Hpt cells were infected with WSSV (MOI = 2) and collected at 12 hpi for RNA extraction. The dsRNA-targeted GFP was used as negative control.

### 2.6. Pharmacological Treatment and Cytotoxicity Test

The Hpt cell cultures were seeded into 24- or 96-well culture plates and cultured in 500 μL or 100 μL of L15 medium (HyClone, Logan, UT, USA) with minor modifications. Hpt cells were incubated with the SIRT1 activator CAY10602 (MedChemExpress, Princeton, NJ, USA) for 12 h at 20 °C. Cell viability following drug treatment was assessed with an MTT (3-(4,5-dimethylthiazol-2-yl)-2, 5-diphenyltetrazolium bromide) (Sigma-Aldrich, Saint Louis, MO, USA) assay according to the manufacturer’s protocol. Hpt cells were pre-treated with the indicated concentration for 12 h followed by infection with WSSV (MOI = 2 for 24-well plates or MOI = 5 for 96-well plates), and then they were collected at indicated time points for RNA extraction or Western blot analysis.

### 2.7. Prokaryotic Expression and Purification of Recombinant Protein

*Cq*SIRT1 recombinant protein fused with His tag was expressed from pET-32a (+) in *E. coli* BL21 (DE3), which were both induced with 0.1 mM isopropyl-1-thio-β-D-galactopyranoside (IPTG) for 20 h at 16 °C. Then, the His-tagged *Cq*SIRT1 recombinant protein was purified with a Ni-NTA affinity column (Smart Lifesciences, Changzhou, China) and washed with washing buffer (50 mM Tris-HCl, 0.5 M NaCl, 25 mM imidazole; pH 8.0). Finally, the purified recombinant protein was collected with an elution buffer (50 mM Tris-HCl, 500 mM NaCl, 250 mM imidazole; pH 8.0) and dialyzed with buffer solution (50 mM Tris, 150 mM NaCl, 5% glycerol). The GST-tagged VP28 recombinant prokaryotic expression vector was constructed previously [14]. The GST-VP24 and GST-VP26 vectors were generated with PCR and subcloned into a pGEX-4T-1 vector between the *B*amH I and *Eco*R I restriction sites. These recombinant proteins were also induced with 0.1 mM IPTG and purified with glutathione beads (Smart Lifesciences, Changzhou, China), which was followed by elution with L-glutathione reduced solution (Solarbio, Beijing, China) (50 mM Tris-HCl, 20 mM L-glutathione reduced, 5 mM DTT; pH = 8.0). These recombinant proteins were stained with Coomassie brilliant blue and analyzed in 12% SDS-PAGE.

### 2.8. Pull-Down Assay

One hundred micrograms of GST or GST-tagged recombinant protein mixed with 30 μL of glutathione beads was incubated with end-over-end mixing for 1 h at 4 °C. Then, the supernatant was removed after centrifugation at 500× *g* for 1 min at 4 °C. Two hundred and fifty micrograms of His-tagged *Cq*SIRT1 recombinant protein was added into the glutathione beads for incubation with end-over-end mixing for 4 h at 4 °C. The beads were then washed with buffer solution (50 mM Tris, 150 mM NaCl, 5% glycerol) five times and eluted with 30 μL of L-glutathione reduced buffer (50 mM Tris-HCl, 20 mM L-glutathione reduced, 5 mM DTT; pH = 8.0). The protein samples were stained with Coomassie brilliant blue or subjected to Western blot analysis.

### 2.9. Transfections and Co-Immunoprecipitation

Flag-*Cq*SIRT1 plasmid was generated with PCR and subcloned into the pxj40-Flag vector (Invitrogen, Waltham, MA, USA) between the *Xho* I and *B*amH I restriction sites. HA-VP24, HA-VP26 and HA-VP28 plasmids were generated with PCR and subcloned into the pCMV-HA vector (Invitrogen, Waltham, MA, USA) between the *Eco*R I and *Xho* I restriction sites. Plasmid transient transfection was performed using Lipo293™ Transfection Reagent according to the manufacturer’s protocol (Beyotime Biotechnology, Shanghai, China). Whole-cell extracts were generated through direct lysis with Western and IP lysis buffer (20 mM Tris (pH 7.5), 150 mM NaCl, 1% Triton X-100) supplemented with protease inhibitor cocktail (EDTA-free, 100× in DMSO) (ApexBio Technology, Houston, TX, USA). Lysates were collected through centrifugation at 13,000× *g* for 10 min at 4 °C. Immunoprecipitation was carried out by incubating Magnetic Beads-Conjugated Mouse Anti Flag-Tag mAb at 4 °C with lysate for 2 h (ABclonal Technology, Wuhan, China), which were then washed three times with cold Western and IP lysis buffer and eluted with 1 × SDS loading buffer (Solarbio, Beijing, China).

### 2.10. SDS-PAGE and Western Blot Analysis

The protein samples were boiled with 1 × SDS or 5 × SDS sample loading buffer for 10 min and analyzed with 12% sodium dodecyl sulfate polyacrylamide gel electrophoresis (SDS-PAGE). After being transferred to PVDF membranes (Merck Millipore, Billerica, MA, USA), the membranes were blocked in 5% non-fat milk in 1 × Tris-buffered saline with 0.1% Tween 20 (TBST) and then incubated with diluted anti-His monoclonal antibody (SAB, Fairfield, NJ, USA), anti-VP28 monoclonal antibody, anti-β-actin monoclonal antibody (TransGen Biotech, Beijing, China), anti-FLAG monoclonal antibody (ABclonal Technology, Wuhan, China) or anti-HA polyclonal antibody (Proteintech Group, Chicago, IL, USA) at room temperature for 1 h, respectively. After washing with TBST three times, goat anti-mouse IgG monoclonal antibody conjugated to horseradish peroxidase or goat anti-rabbit IgG polyclonal antibody conjugated to horseradish peroxidase (Thermo Fisher Scientific, Waltham, MA, USA) were used as secondary antibodies. After a final wash with TBST three times, the membrane was exposed using an Enhanced Chemiluminescent Kit (NCM Biotech, Suzhou, China).

### 2.11. Immunofluorescence Assay

The Hpt cell cultures were seeded into cover slides and cultured in 200 μL of L15 medium. After treatment with drugs, Hpt cells were infected with WSSV (MOI = 100) and incubated for 24 h at 26 °C. Culture medium was removed from cover slides and washed with CPBS to clear residual cell debris. Cells were fixed with 4% paraformaldehyde for 10 min at 4 °C and permeated with TritonX-100 for 30 min at room temperature. Subsequently, the cells were blocked for 1 h at room temperature with 5% goat serum dissolved in CPBS. After washing with CPBS, the cells were incubated with anti-SIRT1 polyclonal antibody (ABclonal, Wuhan, China) and anti-VP28 monoclonal antibody at a dilution of 1:300 with CPBS for 2 h at room temperature. The cells were then rinsed with CPBS three times and incubated with Alexa Fluor488 conjugated goat anti rabbit IgG (H + L) and Alexa Fluor 594 conjugated goat anti mouse IgG (H + L) (ZS-Bio, Beijing, China) at a dilution of 1:500 with CPBS for 2 h at room temperature. After a final wash, the cell nuclei were stained with 4′, 6′-diamidino-2′-phenylindole (DAPI) (Beyotime Biotechnology, Shanghai, China) for 10 min at room temperature and analyzed under a confocal laser scanning microscope (Carl Zeiss, Jena, Germany).

### 2.12. Statistical Analysis

The data was analyzed using Student’s *t*-test and assessed as the mean ± SD from three independent assays for comparison between two groups using GraphPad Prism, in which *p* < 0.05 was considered as statistically significant.

## 3. Results

### 3.1. Molecular Characterization of CqSIRT1 from Red Claw Crayfish C. quadricarinatus

In order to explore the role of *Cq*SIRT1 in WSSV infection, *Cq*SIRT1 was initially identified and analyzed from red claw crayfish (GenBank: MT024786.1). The open reading frame (ORF) of *Cq*SIRT1 was 2256 bp and encoded a 751 amino acid protein with a predicted molecular weight of 83.7 kDa. Conserved domain analysis showed that *Cq*SIRT1 contained a typical conserved catalytic core domain (240 to 472 amino acids) with deacetylase activity and highly variable N- and C-terminal domains (Figure 1A), suggesting that *Cq*SIRT1 may serve as a conserved lysine deacetylase. Accordingly, both the N and C terminal regions help the direct catalytic core domain (about 250 amino acids) for a variety of targets. Moreover, *Cq*SIRT1 exhibited 90% complete amino acid sequence identity to SIRT1 from the crayfish *Procambarus clarkia* and 76% complete amino acid sequence identity to SIRT1 from prawn, such as *Penaeus vannamei*, *Penaeus monodon* and *Penaeus chinensis*, but relatively lower identity to SIRT1 homologs from vertebrates, such as *Danio rerio*, *Mus musculus* and *Xenopus laevis*. Multiple sequence alignment based on the conserved catalytic core domain showed highly identical SIRT1 homologs from other species (Figure 1B). Phylogenetic analysis further revealed that *Cq*SIRT1, with homologs from several invertebrates, was clearly presented as an independent branch (Figure 1C). Together, these data suggest that *Cq*SIRT1 is an evolutionarily conserved lysine deacetylase.

### 3.2. Tissue Distribution and the Effect of CqSIRT1 on WSSV Gene Transcription

To analyze the expression level of the *Cq*SIRT1 transcript in different tissues or organs of red claw crayfish, the transcript of *Cq*SIRT1 was determined using semi-quantitative RT-PCR in various tissues or organs, including the Hpt, haemocyte, gill, stomach, heart, hepatopancreas, epithelium, gonad, muscle, nerve, intestine and eyestalk from red claw crayfish. As shown in Figure 2A, the transcripts of *Cq*SIRT1 were ubiquitously expressed in all examined tissues, with the highest expression in the Hpt, hemocyte, and gill. To further explore how *Cq*SIRT1 functions in WSSV infection, the transcript of the late viral gene *vp28* was determined in Hpt cells after gene knockdown of *Cq*SIRT1 followed by WSSV infection. As shown in Figure 2B, the gene knockdown of *Cq*SIRT1 was nearly 80% more efficient in Hpt cells transfected with *Cq*SIRT1 dsRNA compared to the Hpt cells transfected with GFP dsRNA. Moreover, the transcript of the late viral gene *vp28* was significantly decreased at 12 h post-WSSV infection in Hpt cells after gene knockdown of *Cq*SIRT1 compared to that of negative controls (Figure 2C), indicating that *Cq*SIRT1 plays a positive role during WSSV infection.

### 3.3. Enhancement of Deacetylase Activity of CqSIRT1 Promotes WSSV Replication

To determine whether the impact of *Cq*SIRT1 on WSSV infection was directly driven by deacetylase activity, the effect of CAY10602, a known SIRT1 activator, on viral gene transcription was evaluated through pre-treatment in Hpt cells followed by WSSV infection. First, a concentration gradient of CAY10602 was tested to determine the cell viability of Hpt cells. As shown in Figure 3A, the cell viability of Hpt cells was significantly reduced when CAY10602 concentrations of 100 μM and 200 μM were used, while 50 μM of CAY10602 did not affect cell viability, suggesting that concentrations lower than 50 μM did not affect the viability of Hpt cells. Furthermore, both the transcripts of the late viral envelope gene *vp28* and the nucleocapsid gene *vp664* were markedly increased in Hpt cells if pre-treated with CAY10602 in a dose-dependent manner and followed by WSSV infection, indicating that the deacetylation activity of *Cq*SIRT1 could promote WSSV gene transcription (Figure 3B).

To further examine which stage of the WSSV life cycle was regulated by the deacetylase activity of *Cq*SIRT1, the representatives of different stages of WSSV infection, including the immediately early gene *ie1*, the later viral envelope gene *vp28* and the later viral nucleocapsid gene *vp664*, in Hpt cells following WSSV infection at different time points were analyzed. As shown in Figure 3C–E, the transcript of the *ie1* gene in the CAY10602 treatment group was significantly increased compared to that of the DMSO treatment group at 3 h post-WSSV infection, suggesting that deacetylase activity of *Cq*SIRT1 is essential to WSSV infection at the early stage. Moreover, the transcripts of both *vp28* and *vp664* were markedly increased at 6 hpi and 12 hpi, further indicating that *Cq*SIRT1 driven by deacetylase activity plays an important role during WSSV infection. To prove the effect of the drug on the production of progeny virions, viral genome copies were further analyzed in Hpt cells at 24 h and 48 h post-WSSV infection, respectively. Consistently with the above findings at the transcript level, viral genome copies were also notably enhanced in Hpt cells after pre-treatment with CAY10602 compared to those of the treatment with DMSO (Figure 3F). In sum, the effect of *Cq*SIRT1 on the facilitation of WSSV replication was strongly dependent on its deacetylase activity.

### 3.4. Enhancement of Deacetylase Activity of CqSIRT1 Facilitates VP28 Protein Expression

To further prove the role of deacetylase activity driven by *Cq*SIRT1, the effect of promoting WSSV infection was further confirmed by monitoring the protein level of viral envelope protein VP28 using Western blot analysis and an immunofluorescence assay. As expected, viral envelope protein VP28 was observably increased in Hpt cells after pre-treatment with CAY10602 followed by WSSV infection at 12 hpi in a dose-dependent manner (Figure 4A), which was consistent with the transcription level of *vp28* in Hpt cells after pre-treatment with CAY10602. These effects of the pharmacological treatment on WSSV replication were further proved with an immunofluorescence assay performed with a laser scanning confocal microscope. Obviously, the fluorescence signal of VP28 protein was strongly aggregated in the nuclei of Hpt cells treated with CAY10602 followed by WSSV infection at 24 hpi when compared to that of the DMSO control (Figure 4B,C), suggesting that the deacetylase activity of *Cq*SIRT1 markedly promoted production of viral VP28 proteins. These findings further indicate that the deacetylase activity of *Cq*SIRT1 is essential to WSSV infection.

### 3.5. CqSIRT1 Co-localizes with VP28 in the Nuclei of Hpt Cells and Binds with VP28

To further reveal the relationship between *Cq*SIRT1 and WSSV, as well as the molecular mechanism underlying the impact of *Cq*SIRT1 on WSSV infection, localization of *Cq*SIRT1 in Hpt cells utilizing a specific antibody towards WSSV infection was investigated. Similar to SIRT1 homologs of vertebrates, *Cq*SIRT1 was predominantly distributed in the nuclei of Hpt cells (Figure 5A). Surprisingly, co-localization of *Cq*SIRT1 with VP28 in the nuclei of Hpt cells at 12 hpi and 24 hpi was markedly observed under a laser confocal microscope, seeming to result in a *Cq*SIRT1 aggregate when exposed to WSSV infection (Figure 5A). To determine whether *Cq*SIRT1 could directly bind to VP28, recombinant protein pulldown assays were evaluated in vitro. First, *Cq*SIRT1 recombinant protein fused with His tag and VP28 recombinant protein fused with GST tag were expressed and purified from *E. coli*. Then, the GST pulldown assay between *Cq*SIRT1 and VP28 recombinant protein was performed, and this was followed by Western blot analysis. As shown in Figure 5B, His-*Cq*SIRT1 recombinant protein was found to be able to directly bind to GST-VP28 recombinant protein in the Western blot analysis with anti-His monoclonal antibody. To further prove the above result, Flag-tagged *Cq*SIRT1 and HA-tagged VP28 were co-transfected into HEK293T cell lines, and then co-immunoprecipitation analysis was performed. As shown in Figure 5C, Flag-*Cq*SIRT1 was consistently found to bind with HA-VP28 in the Western blot analysis. These results demonstrate that *Cq*SIRT1 could specifically directly bind with envelope protein VP28.

### 3.6. CqSIRT1 also Binds with Viral Envelope Proteins VP24 and VP26

Since the viral envelope proteins VP28, VP24 and VP26 are abundant proteins that play vital roles during WSSV infection, we examined whether the other two envelope proteins, VP24 and VP26, could also bind to *Cq*SIRT1. First, GST-tagged VP24 and VP26 recombinant protein were expressed and purified from *E. coli*. The pulldown assay between His-*Cq*SIRT1 recombinant protein and GST-VP24 or GST-VP26 recombinant protein was then performed. As shown in Figure 6A,B, *Cq*SIRT1 recombinant protein was also found to be directly bound to both VP24 and VP26 recombinant protein through Coomassie brilliant blue staining and Western blot analysis. To further prove the above result, Flag-*Cq*SIRT1 and HA-VP24 or HA-VP26 were co-transfected into HEK293T cell lines, which was followed by co-immunoprecipitation analysis. As shown in Figure 6C,D, Flag-*Cq*SIRT1 was also found to bind with HA-VP24 or HA-VP26 in the Western blot analysis. In sum, *Cq*SIRT1 could not only bind to VP28 but also physically bound to other two key viral envelope proteins, VP24 and VP26, raising the possibility that *Cq*SIRT1 may precisely modulate the functions of partial viral structural proteins via their interactions with each other during WSSV infection in a deacetylase activity-dependent manner. However, the potential molecular mechanism underlying the relationship between *Cq*SIRT1 and viral structural proteins still needs to be further explored.

## 4. Discussion

A growing body of studies highlights protein lysine acetylation as playing a vital regulatory role in viral pathogenicity and the host immune defense system. Along with lysine acetylases, lysine deacetylases responsible for substrates lysine deacetylation modulate the activity of proteins and their cellular functions [20]. SIRT1 is the best-characterized, showing powerful deacetylase activity, and it is strongly associated with longevity and diseases caused by virus infection [21]. In our present study, *Cq*SIRT1 was functionally characterized from red claw crayfish *C. quadricarinatus* during WSSV infection. Multiple sequence alignment of the catalytic core domain within *Cq*SIRT1 revealed that it was highly identical to that of homologs from other species, which indicated that *Cq*SIRT1 may also possess conserved deacetylase activity. Thus, CAY10602, the known SIRT1 activator applied in human disease research, makes functional studies associated with deacetylase activity possible in crustaceans. Our study found that enhancement of the deacetylase activity of *Cq*SIRT1 significantly promoted WSSV infection. Moreover, *Cq*SIRT1 could bind to several viral envelope proteins, such as VP24, VP26 and VP28, which is rarely reported for other viruses. Those viral envelope proteins are the most abundant proteins in WSSV, among which VP26 and VP28 account for approximately 60% of the total number of envelope proteins [22,23,24,25]. Whether binding of *Cq*SIRT1 to the viral envelope protein is required for the viral life cycle still needs to be further investigated.

Previously, several studies proved that SIRT1 plays a vital role in various virus infections by binding to viral trans-activator or non-structural proteins with enzymatic activity. For instance, SIRT1 was recruited to viral E1-E2-replicating DNA via interaction with viral helicase E1 and encoding protein E2, both of which are associated with transcriptional regulation and initiation of the viral genome, thereby activating the deacetylation enzyme activity of SIRT1 and deacetylating the DNA repair factor Werner helicase to promote the interaction with E1-E2-replicating DNA [26]. In the case of KSHV infection, SIRT1 regulated KSHV latency by binding to the viral master lytic transactivator RTA promoter and inhibiting RTA transactivation, as well as downstream target gene expression, such as viral interleukin-6 [5]. In fact, a previous study found that an SIRT1 homolog identified from white shrimp (*Litopenaeus vannamei*) could facilitate the promoter activity of the immediately early viral gene ie1 in WSSV, which indicates that SIRT1 could also regulate viral infection through modulation of gene transcriptional machinery in crustaceans [15]. To the best of our knowledge, the finding that SIRT1 targets viral structural proteins has not been reported before. Therefore, the finding concerning the interaction between *Cq*SIRT1 and viral envelope protein may reveal a novel mechanism by which SIRT1 plays a modulatory role during viral infection through targeting of viral structural proteins.

In fact, although a great many studies have found that HDAC family deacetylases are recruited to viral protein-associated complexes and the viral genome, targeting of viral structural proteins for other classes of deacetylases is rarely reported. For example, in the case of Herpes simplex virus-1 infection, HDAC1 could interact with viral ICP0, ICP8 and US3 for redistribution of the HDAC1 complex to the cytoplasm, localization of HDAC1/ICP0 to ND10 bodies or promotion of HDAC1 phosphorylation, respectively [27,28,29]. Moreover, as key chromatin-associated transcriptional regulators, HDACs exhibit a powerful function by promoting compact chromatin organization and, thereby, modulating gene transcription [30,31]. In the present study, we mainly focused on the relationship between *Cq*SIRT1 and viral structural proteins, which is a novel approach for better understanding the potential role of *Cq*SIRT1 during virus infection. Since *Cq*SIRT1 was co-localized with VP28 in the nuclei of Hpt cells at late stage and *Cq*SIRT1 could also bind to two other key viral envelope proteins, VP24 and VP26, combining these findings with the effect of *Cq*SIRT1 driven by deacetylase activity on WSSV infection, we speculate that *Cq*SIRT1 may participate in regulating partial viral structural proteins in a deacetylase activity-dependent way. However, more details are necessary to further elucidate this process.

In conclusion, *Cq*SIRT1 was characterized from *C. quadricarinatus* with conserved lysine deacetylase activity, and both RNA interference and pharmacological treatment showed that *Cq*SIRT1 could facilitate WSSV replication. More importantly, *Cq*SIRT1 could bind to several viral envelope proteins, such as VP24, VP26 and VP28, as shown by the GST pulldown assay and co-immunoprecipitation. This study provides the novel finding that *Cq*SIRT1 may modulate viral infection through linkage to viral structural proteins and expands our knowledge about the effect of lysine deacetylases on pathogenic mechanisms caused by viruses.

## Figures and Tables

**Figure 1 viruses-14-01733-f001:**
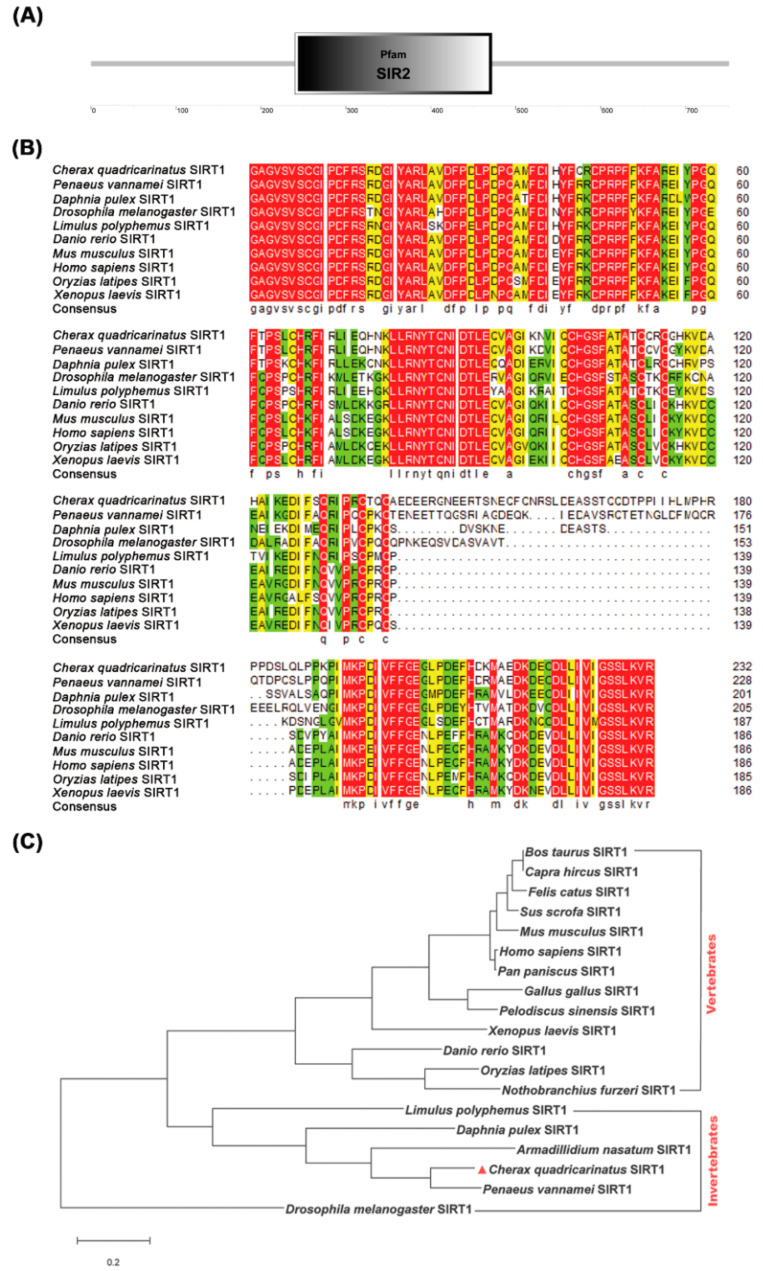
Molecular characteristics of *Cq*SIRT1 from red claw crayfish *C. quadricarinatus*. (**A**) Conserved domain analysis of *Cq*SIRT1. (**B**) Multiple sequence alignment of *Cq*SIRT1 based on deacetylase domains. (**C**) The phylogenetic tree of *Cq*SIRT1 homologs from several species based on complete amino acid sequences using the neighbor-joining method. *Penaeus vannamei* (XP_027218224.1), *Armadillidium nasatum* (KAB7503239.1), *Daphnia pulex* (XP_013777128.1), *Limulus Polyphemus* (XP_013777128.1), *Drosophila melanogaster* (AAF53248.1)*, Danio rerio* (XP_001334440.4), *Mus musculus* (AAI52315.1), *Sus scrofa* (NP_001139222.1), *Bos Taurus* (NP_001179909.3), *Felis catus* (NP_001277175.1), *Gallus gallus* (NP_001004767.1), *Pelodiscus sinensis* (XP_006125338.1), *Pan paniscus* (XP_003830353.1), *Oryzias latipes* (XP_004077552.1), *Xenopus laevis* (NP_001091195.1), *Capra hircus* (AKJ66821.1), *Nothobranchius furzeri* (ABX71822.1), *Homo sapiens* (AAH12499.1).

**Figure 2 viruses-14-01733-f002:**
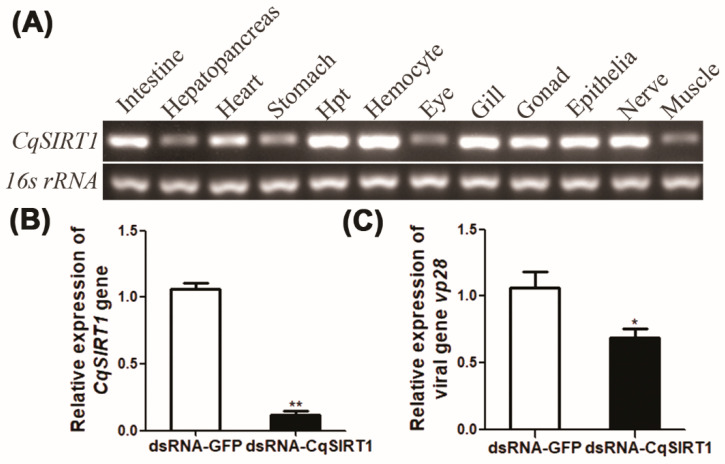
Tissue distribution and the effect of *Cq*SIRT1 on WSSV gene transcription. (**A**) The relative expression of the *CqSIRT1* gene in different tissues from *C. quadricarinatus* as shown by semi-quantitative RT-PCR. The 16s ribosome RNA served as the inter-control. (**B**) Gene knockdown efficiency of *Cq*SIRT1 in Hpt cells. The relative expression of *Cq*SIRT1 was determined in Hpt cells transfected with *Cq*SIRT1 dsRNA followed by WSSV infection. The GFP dsRNA was used as negative control. (**C**) The relative expression of late viral gene *vp28* in *Cq*SIRT1-silenced Hpt cells. The relative expression of *vp28* was measured in Hpt cells transfected with *Cq*SIRT1 dsRNA followed by WSSV infection. The GFP dsRNA served as negative control. These experiments were performed in biological triplicates (*, *p* < 0.05, **, *p* < 0.01).

**Figure 3 viruses-14-01733-f003:**
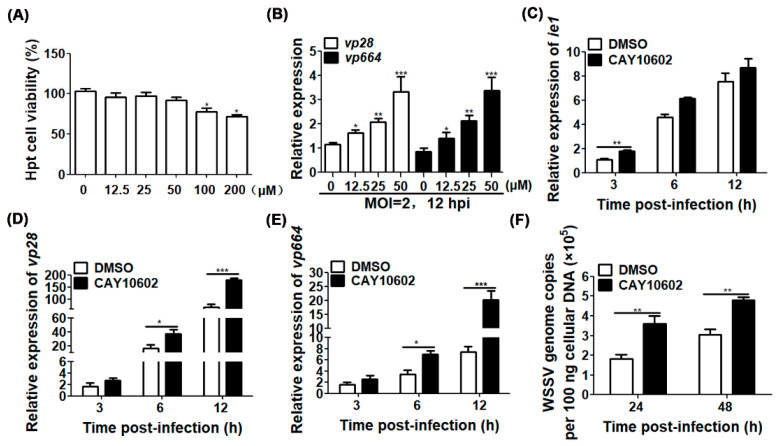
Enhancement of *Cq*SIRT1 deacetylase activity promoted WSSV replication. (**A**) Cytotoxicity test of SIRT1 activator CAY10602 in Hpt cells. Cell viability was evaluated in Hpt cells pre-treated with the indicated concentration of CAY10602 using the MTT assay. (**B**) The relative expressions of *vp28* and *vp664* in Hpt cells treated with CAY10602 in a dose-dependent manner. The relative expressions of both *vp28* and *vp664* were determined in Hpt cells pre-treated with the indicated concentration of CAY10602 followed by WSSV infection (MOI = 2). DMSO was used as negative control. (**C–E**) The relative expressions of viral genes in Hpt cells treated with CAY10602 at different stages during WSSV infection. The relative expressions of *ie1*, *vp28* and *vp664* genes were determined in Hpt cells treated with 50 μM of CAY10602 followed by WSSV infection for the indicated time points. Treatment with DMSO served as negative control. (**F**) WSSV genome copies in Hpt cells treated with CAY10602. The viral genome copies were determined in Hpt cells treated with 50 μM of CAY10602 followed by WSSV infection for 24 h and 48 h. DMSO served as control treatment. All experiments were performed in biological triplicates and means ± SD were shown (*, *p* < 0.05; **, *p* < 0.01, ***, *p* < 0.001).

**Figure 4 viruses-14-01733-f004:**
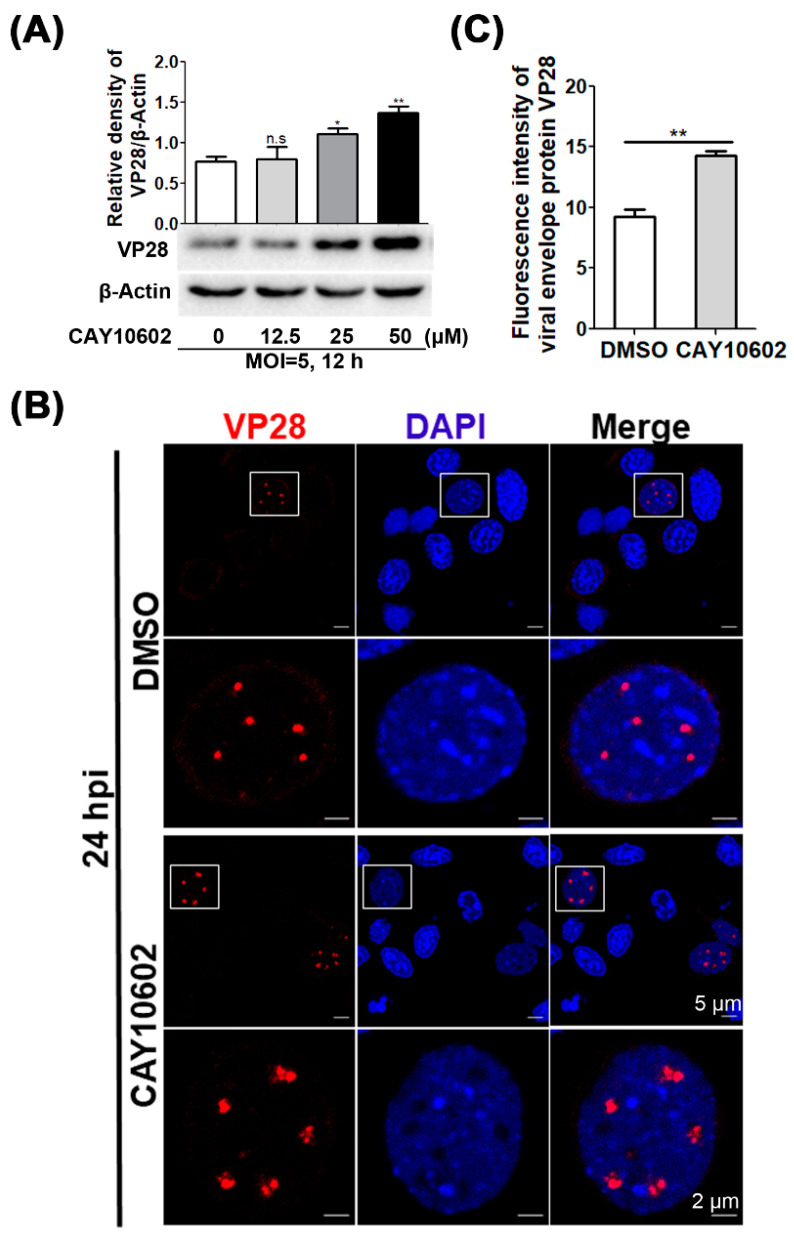
Enhancement of *Cq*SIRT1 deacetylase activity promoted VP28 protein expression. (**A**) VP28 protein was significantly increased in Hpt cells through the activation of deacetylase with the activator CAY10602. VP28 protein level was determined using Western blot analysis. β-actin was used as an inter-control. The relative quantitation of VP28 protein expression was conducted with histogram analysis. (**B**) VP28 protein was strongly aggregated in the nuclei of Hpt cells treated with CAY10602. The fluorescence signal of VP28 (red) was observed under a laser confocal microscope. (**C**) The relative quantitation of the fluorescence intensity of VP28 was determined with histogram analysis. These experiments were performed in biological triplicates (*, *p* < 0.05, **, *p* < 0.01).

**Figure 5 viruses-14-01733-f005:**
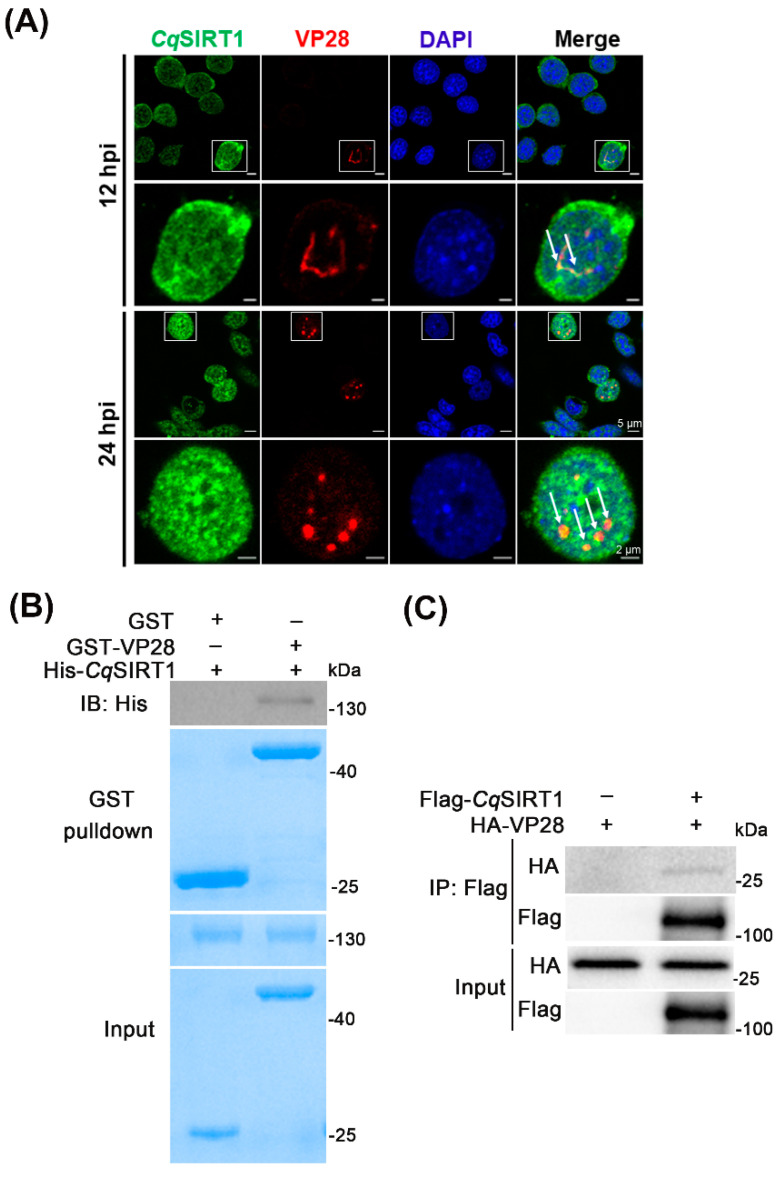
*Cq*SIRT1 was co-localized with and bound with VP28. (**A**) *Cq*SIRT1 was co-localized with VP28 in the nuclei of Hpt cells under a laser confocal microscope using specific antibodies against *Cq*SIRT1 (green) and VP28 (red). The arrow indicates co-localization. (**B**) GST pulldown assay between *Cq*SIRT1 and VP28 recombinant protein. The GST pulldown assay was performed, followed by Western blot analysis with anti-His monoclonal antibody. GST recombinant protein served as negative control. (**C**) Flag-*Cq*SIRT1 bound to HA-VP28 in the co-immunoprecipitation assay. Flag-*Cq*SIRT1 and HA-VP28 plasmids were co-transfected into HEK293T cell lines, then subjected to co-immunoprecipitation and Western blot analysis with the indicated specific antibody. These experiments were performed in biological triplicates.

**Figure 6 viruses-14-01733-f006:**
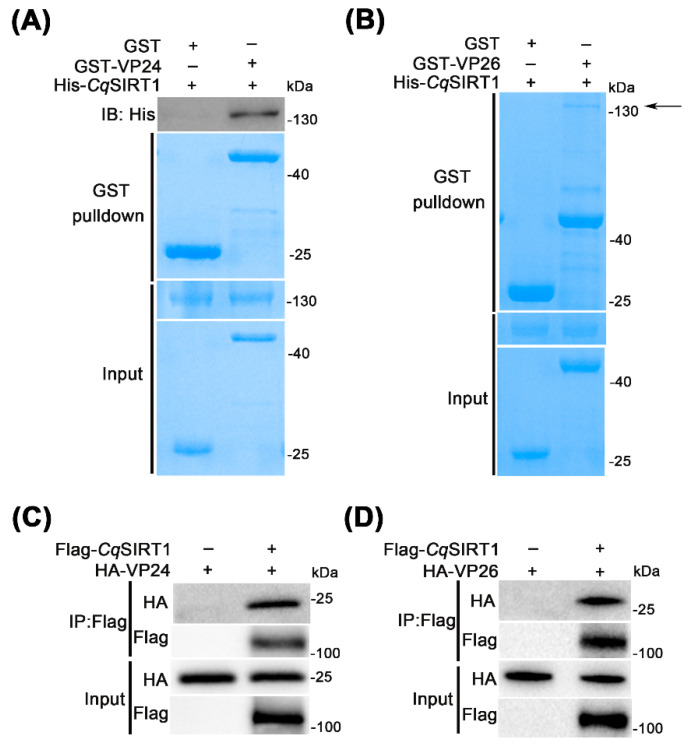
*Cq*SIRT1 bound to viral envelope proteins VP24 and VP26. (**A**,**B**) GST pulldown assay between recombinant protein *Cq*SIRT1 and VP24 or VP26. The GST pulldown assay was performed and followed by Coomassie brilliant blue staining and Western blot analysis with anti-His monoclonal antibody. GST recombinant protein served as negative control. (**C**,**D**) Flag-*Cq*SIRT1 was found to bind to HA-VP24 or HA-VP26 in the co-immunoprecipitation assay. Flag-*Cq*SIRT1 and HA-VP24 or HA-VP26 plasmids were co-transfected into HEK293T cell lines, then subjected to co-immunoprecipitation and Western blot analysis with the indicated specific antibody. These experiments were performed in biological triplicates.

**Table 1 viruses-14-01733-t001:** Primer sequences used in this study.

Primer	Sequence (5′–3′)	Usage
SIRT1-F	ATGGCGGACGCAGCATACGGG	ORF
SIRT1-R	TCATAGGTCAGATGAAGAAC	
qSIRT1-F	CCCAGACCCCTTCCATCAAC	qRT-PCR
qSIRT1-R	GAGGCAACAGTCTCCCGAAT	
16s rRNA-F	AATGGTTGGACGAGAAGGAA	qRT-PCR
16s rRNA-R	CCAACTAAACACCCTGCTGATA	
qvp28-F	AAACCTCCGCATTCCTGT	qRT-PCR
qvp28-R	GTGCCAACTTCATCCTCATC	
qIE1-F	CTGGCACAACAACAGACCCTACC	qRT-PCR
qIE1-R	GGCTAGCGAAGTAAAATATCCCCC	
VP664(13785+)	TTCTACTGTTGTCGGTCGCC	qRT-PCR
VP664(13940−)	GCGTCTCTATTGATGCGGGA	
dsSIRT1-F	AGGAAGAGGAAGAGGATG	RNAi
dsSIRT1-R	ATTTGAAGAAGGGACGAG	
dsGFP-F	CGACGTAAACGGCCACAAGT	RNAi
dsGFP-R	TTCTTGTACAGCTCGTCCATGC	
SIRT1(32a)-F	CGGGATCCATGGCGGACGCAGCATACGG	Prokaryoticexpression
SIRT1(32a)-R	CCGCTCGAGTAGGTCAGATGAAGAACATT	
VP24-F	GAGAGGATCCACCAACATAGAACTTAAC	
VP24-R	GAGAGAATTCTTTTTCCCCAACCTTAAAC	
VP26-F	GAGAGGATCCATGACACGTGTTGGAAG	
VP26-F	GAGAGAATTCCTTCTTCTTGATTTCGTC	
Flag-S1-F	CGGGATCCATGGCGGACGCAGCATACGG	Eukaryotic expression
Flag-S1-R	CCGCTCGAGTCATAGGTCAGATGAAGAACATT	
HA-VP24-F	CCGGAATTCGGATGCACATGTGGGGGGTTTAC	
HA-VP24-R	CCGCTCGAGTTATTTTTCCCCAACCTTAAAC	
HA-VP26-F	CCGGAATTCGGATGGAATTTGGCAACCTAAC	
HA-VP26-R	CCGCTCGAGTTACTTCTTCTTGATTTCGT	
HA-VP28-F	CCGGAATTCGGATGGATCTTTCTTTCACTC	
HA-VP28-R	CCGCTCGAGTTA CGCGGATCCCTCGGTCTC	

Underlined text indicates the restriction enzyme cutting sites *Eco*R I (GAATCC), *Xho* I (CTCGAG) and *Bam*H I (GGATCC).

## Data Availability

Not applicable.

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
