# Peer review of "A Deacetylase *Cq*SIRT1 Promotes WSSV Infection by Binding to Viral Envelope Proteins in *Cherax quadricarinatus"

_viruses, 2022, doi:10.3390/v14081733_

Round 1
Reviewer 1 Report
In this manuscript, the authors demonstrated A deacetylase CqSIRT1 promotes WSSV infection by binding to viral envelope proteins in Cherax quadricarinatus. This manuscript is well-written and contained enough results for viruses. However, there are still several issues to be corrected or improved as follows:
1. Section 2.1, details about the health of C. quadricarinatus, if they were fed/frequency, Hpt cell cultures and WSSV are missing.
2. Section 2.3, the authors neglect to provide citations for the manuscripts that created the bioinformatics tools.
3. Section 2.7, The author states that GST-tagged VP24, VP26 or VP28 have been studied in the previous study. If this part of the results is published, it is necessary to quote. Otherwise, detailed experimental steps and results should be written in this manuscript.
4. Figure 2A, Why CqSIRT1 was highest expression in Hpt, hemocyte, and gill, is this the case in other species, and is there a relationship with CqSIRT1 function. The font of 16s rRNA should be italic. The font of ‘p’ in ‘*, p<0.05; **, p<0.01’ should be italic.
5. The MOI of WSSV was different in Figure 3B and Figure 4A, why did the authors infect cells with different viral titers.
6. Figure 4B, VP28 is not detected in every cell, what causes this situation? Normally, all cells would be infected, and VP28 would be detected in all cells.
Reviewer 2 Report
In the manuscript, SIRT1 was cloned and characterized from red claw crayfish Cherax quadricarinatus. Gene knockdown of CqSIRT1 in crayfish Hpt cell culture using RNAi inhibited WSSV late envelope gene vp28 transcription, while enhancement of its deacetylase activity using pharmacological activator promoted vp28 expression as well as viral replication. Furthermore, CqSIRT1 was colocalized with viral envelope protein VP28 in the nuclei of Hpt cell and directly bound to VP28 by pull down and co-immunoprecipitation. CqSIRT1 was also interacted with another two viral envelope proteins. Overall, it can be a valuable contribution to the field. However, several points require attention and should be addressed as described below.
1. line 26-27 should be by pull-down and co-immunoprecipitation assays.
2. line 87 please insert reference for Hpt cell isolation and culture.
3. line 102 performed should be replaced with cloned.
4. line 113 MAGA X? should be MEGA.
5. line 166, 188 please insert reference for expression vectors.
6. line 217 what's the full name of CPBS?
7. line 244-248 There is no any information about SIRT1 sequence from Procambarus clarkia, Penaeus monodon and Penaeus chinensis in Figure 1B, 1C.
8. line 252, 253 shift (Figure.1C) from line 253 to line 252.
9. line 280 please detect WSSV genome copy after knockdown of CqSIRT1 in Figure 2.
10. line 377 In Figure 5A, the cell in the top panel looks so longer than most cells. Different cell type? Please use the most common cell type.
11. line 395 remove an extra period.
12. line 405 why not detect His-CqSIRT1 using WB?
